# Transcriptome Analysis on the Mechanism of Ethylicin Inhibiting *Pseudomonas syringae* pv. *actinidiae* on Kiwifruit

**DOI:** 10.3390/microorganisms9040724

**Published:** 2021-03-31

**Authors:** Tao Liu, Xiaoli Ren, Guangyun Cao, Xia Zhou, Linhong Jin

**Affiliations:** State Key Laboratory Breeding Base of Green Pesticide and Agricultural Bioengineering, Key Laboratory of Green Pesticide and Agricultural Bioengineering, Ministry of Education, Guizhou University, Guiyang 550025, China; gs.taoliu18@gzu.edu.cn (T.L.); gs.xlren20@gzu.edu.cn (X.R.); gs.gycao20@gzu.edu.cn (G.C.)

**Keywords:** kiwifruit bacterial canker, *Pseudomonas syringae* pv. *actinidiae*, ethylicin, antibacterial action, transcriptome, action mechanism

## Abstract

Bacterial canker disease caused by *Pseudomonas syringae* pv. *actinidiae* (Psa) is a devastating disease of kiwifruit, which is severely limiting the development of the kiwifruit industry. Ethylicin is a broad-spectrum plant biomimetic fungicide. However, its application in the control of kiwifruit bacterial canker is rarely reported, and the mechanism of ethylicin on Psa remains unknown. In this study, we investigated the effect of ethylicin on Psa in vitro and in vivo and found that ethylicin can inhibit the growth of Psa and prevent the cankering in the plant stem. Mechanism investigation indicated that ethylicin acted by limiting the movement of Psa, destroying the cell membrane of Psa, and inhibiting the formation of Psa biofilm. In addition, it was also found through transcriptomics research that ethylicin can up-regulate the expression of genes related to protein export and biofilm formation–*Pseudomonas aeruginosa* and down-regulate the expression of genes related to flagellar assembly in Psa. This study concluded that ethylicin can effectively inhibit Psa growth, and it could help to gain a better understanding of the mechanisms of ethylicin inhibiting Psa and provide practical data for the application of ethylicin as a highly potent agent for controlling the bacterial canker disease of kiwifruit.

## 1. Introduction

Kiwifruit bacterial canker is a devastating disease in the kiwifruit cultivar industry, and *Pseudomonas syringae* pv. *actinidiae* (Psa) is the main pathogen of kiwifruit bacterial canker [1]. In 1980, kiwifruit bacterial canker disease was first reported and identified on kiwifruit in California, USA [2], and was then found in Shizuoka, Japan [3]. The name of the pathogen was determined to be *Pseudomonas syringae* pv. *actinidiae* in 1989 [4]. Now, kiwifruit bacterial canker disease has been found in many main production regions all over the world, including China [5], Italy [6], Iran [7], Portugal [8], and New Zealand [9], and the disease has caused serious yield and economic loss in these countries.

At present, the control of this disease relies on copper-based pesticides and streptomycin [10]. However, the extensive use of copper-based pesticides and antibiotics can lead to the resistance of pathogenic bacteria, changes in the soil bacterial community structure, and environmental pollution [11]. Other bactericides, such as acibenzolar-S-methyl, can reduce the occurrence of bacterial canker, but this was counteracted with a phytotoxicity effect on kiwifruit [12,13]. A sulfur agent was applied but with low efficiency (EC_50_ = 1326.99 mg/L) on Psa, which compromises the concept of highly efficient and safe pesticides [14]. Currently, the control of kiwifruit bacterial canker disease can only rely on prevention rather than a treatment agent [15]. Therefore, potent pesticides with high efficiency on Psa, less side effects, and less environmental pollution are in great need.

Organosulfur compound ethylicin (S-ethyl ethanethiosulfonate; CAS number 682-91-7) is a bionic pesticide that mimics natural allicin obtained from garlic. It was first prepared and studied in the laboratory during the synthetic research of allicin and its homologues in 1958 and developed as a broad-spectrum biomimetic fungicide in China [16]. Ethylicin is under preliminary research for its potential fungicides [17,18,19,20,21]. Moreover, ethylicin can stimulate plant growth, resulting in a faster and robust seed germination [22]. Ethylicin was also found with a high capability against kiwifruit bacterial canker disease caused by [23]. However, the mechanism of ethylicin on Psa has not yet been reported.

The purpose of this study is to evaluate the effect of ethylicin on Psa and analyze its potential mechanism of action through transcriptomics.

## 2. Materials and Methods

### 2.1. Materials

The pesticides used in this experiment were commercially available. Both 80% ethylicin and 12% copper rosinate were obtained from Hainan Zhengye Zhongnong High-tech Co., Ltd., Haikou, Hainan Province, China, and Kocide 3000 was purchased from DuPont de Nemours, Inc., Wilmington, CA, USA. A one-year-old kiwifruit plant (from Hongyang plantation) was used for the in vivo test. The Psa strain was provided by our colleague Dr. Zhibing Wu from Guizhou University. The culture conditions were 28 °C, 180 r/min.

### 2.2. Bioassay Methods

In vitro antibacterial activities of ethylicin against Psa followed the reported methods [24]. Sterile water containing dimethyl sulfoxide (DMSO) was used as a negative check; Kocide 3000 and copper rosinate were used as positive controls. For initial testing of all three agents, the solution concentration was set at 200, 100, and 50 μg/mL, which was incubated with bacterial solution and then procedurally measured for the optical density (OD) value. EC_50_ was further tested at five lower gradient concentrations. Data were collected in triplicate for each agent concentration. Based on the OD value, the inhibitory effect of the bactericide on Psa was calculated.
(1)$$I(\%)=\frac{\left(CK-T\right)}{CK}\times 100\%$$

Equation (1), *I* (%) represents inhibition rate. *CK* is the OD value of the DMSO control group. *T* is the OD value of the bactericide group.

In vivo tests of ethylicin’s efficacy on bacterial canker disease of the one-year-old kiwifruit plant was determined following the method of Liu [25]. The kiwifruit plant branch was cut 1 mm open with a sterilized knife and then injected with 10μL of ethylicin solution at a concentration of 0, 200, and 500 μg/mL. Then, 24 h later, Psa was inoculated to the branch cut. Each treatment was repeated three times. The treatment with DMSO (0 μg/mL of ethylicin) was set as the negative control. Another group with no treatment (no ethylicin and no bacterial inoculation) served as the blank control. Changes in the inoculated branch cut were observed 14 days after inoculation. The control efficiencies *I* (%) for ethylicin were calculated using the following equation:(2)$$I(\%)=\frac{\left(C-T\right)}{C}\times 100$$

Equation (2), *C* and *T* are the average lengths of the lesion of the negative control and the treatment group, respectively. The unit of lesion length is mm.

### 2.3. Time-Concentration Dependence Measurement of Ethylicin on Psa Growth

The effect of ethylicin on the growth of Psa was determined by following the reported method [26]. Psa strains were cultured in nutritive broth medium at 28 °C until OD_600_ = 0.6–0.8. The growth of the cultures was monitored on a microplate reader (Synergy H1, Bio Tek, Winooski, VT, USA) by measuring the optical density at 600 nm (OD_600_), and the turbidity was corrected by subtracting the OD values of the nutrient broth (NB) medium. The cells were harvested and resuspended in an equal volume of sterile water. The 1 mL of Psa bacteria suspension was added to 100 mL of the NB medium containing different concentrations of ethylicin (0, 3.60, 4.50, 5.40, 6.30, 7.2, 8.1, and 9.0 μg/mL). The culture was then incubated at 28 °C, 180 rpm/min, and the value of OD_600_ was measured every 3 h for the following 48 h until bacterial growth reached a stable phase. Each treatment was repeated three times.

### 2.4. Morphological Analysis by Scanning Electron Microscopy (SEM) and Transmission Electron Microscopy (TEM)

The sample preparation procedure for scanning electron microscopy followed the reported method [27]. The 1.5 mL of Psa bacterial solution with OD_600_ = 0.6–0.8 was taken in 2 mL tubes and centrifuged, and the precipitate was washed three times with phosphate buffer solution (PBS) (pH = 7.2) and resuspended in 1.5 mL of PBS. Afterwards, ethylicin was added to make the final concentrations of 0, 50, 75, and 100 μg/mL and incubated at 28 °C and 180 rpm/min for 8 h. These samples were washed three times with PBS (pH = 7.2). Subsequently, the bacterial cells were fixed to dehydrate with 2.5% glutaraldehyde at 4 °C for 12 h and then removed. Next, the samples were dehydrated with ethanol in a gradient in the order of 30%, 50%, 70%, 90%, and 100% ethanol (10 min each time). Finally, the samples were flattened and sprayed with gold and observed with a SEM 450 microscope (FEI, Hillsboro, OR, USA).

The Psa flagella form was then detected by TEM according to the method reported by Zhou [28]. Psa bacterial suspensions (10 μL, OD_600_ = 0.1) were added to an agar plate with nutrient broth medium containing different concentrations (0, 1.8, 3.6, and 5.4 μg/mL) of ethylicin at 28 °C and incubated for 24 h. Then, a piece of bacterial agar was picked using a copper mesh with a carbon support film and then stained with 1% phosphotungstic acid followed by sterile water rinsing. Finally, the treated Psa was observed under transmission electron microscopy (FEI Talos F200C apparatus, Waltham, MA, USA).

### 2.5. Bacteria Swimming

Bacterial motility was measured using a swimming assay according to the method of Lovato [29]. Psa was incubated to OD_600_ values of 0.6–0.8. The 3 μL of bacterial suspension was transferred onto 0.3% agar with nutrient broth medium containing different concentrations of ethylicin (0, 1.80, 3.60, or 7.20 μg/mL). After the plates were incubated at 28°C in the dark for 48 h, bacterial motility was assessed by measuring the diameter of the longest bacterial circles, and each treatment was repeated three times, respectively.

### 2.6. Effect of Ethylicin on Psa Biofilm Formation

According to the method of Ni [30], Psa was incubated to OD_600_ values of 0.6–0.8. Different treatment solutions were mixed with 300 µL of bacterial suspension in 96-well plates to give the final concentrations of ethylicin of 0, 1.80, 2.70, 3.60, and 4.50 μg/mL at 28 °C for 72 h to form biofilms, and each treatment was repeated six times. The bacterial suspensions in the plates were then gently removed, and each well was rinsed three times with sterile water and then dried in a desiccator at 60 °C for 1 h. The biofilms remaining in the 96-well plates were then stained with 0.1% crystal violet solution for 10 min and then rinsed gently three times. Finally, 300 μL of 95% ethanol solution was added to each well, and the solution was subjected to an OD_600_ value measuring to evaluate the content of the formed Psa biofilm.

### 2.7. Psa RNA Extraction, cDNA Library Construction, and Sequencing

A colony of Psa in the NB medium was taken to cultivate to OD_600_ = 0.6–0.8. To the bacterial suspension, the new NB medium and ethylicin was added with the final concentration of ethylicin at 3.60 μg/mL; the same solution with 0 μg/mL ethylicin was set as the control (CK). After cultivation, the solution was centrifuged at 5000 rpm and 4 °C for 5 min; the collected bacteria was washed three times with sterile deionized water and quickly frozen with liquid nitrogen.

Then total RNA was extracted from the above samples with the RNAprep Pure Bacteria Kit (TIANGEN, Carlsbad, CA, USA) by following the manufacturer’s instructions. RNA purity was evaluated using the NanoPhotometer^®^ spectrophotometer (IMPLEN, Westlake Village, CA, USA). RNA integrity was assessed using the RNA Nano 6000 Assay Kit of the Bioanalyzer 2100 system (Agilent Technologies, Santa Clara, CA, USA). Sequencing libraries were generated using the NEBNext^®^ UltraTM RNA Library Prep Kit for Illumina^®^ (NEB, USA) following manufacturer’s recommendations, and index codes were added to attribute sequences to each sample. Library quality was assessed on the Bioanalyzer (Agilent, Santa Clara, CA, USA). The library preparations were sequenced on the NovaSeq 6000 (Illumina, San Diego, CA, USA), and 150 bp paired-end reads were generated.

### 2.8. RNA-Seq Data Analysis

Clean reads were obtained by removing reads containing adapter, reads containing ploy-N, and low-quality reads from raw data. At the same time, Q20, Q30, and GC content of the clean reads were calculated. The clean reads were mapped to the reference genome of *Pseudomonas syringae* pv. *tomato* DC3000 using Bowtie2 v.2.3.4.3 [31]. HTSeq v.0.6.1 was used to count the reads numbers mapped to each gene [32]. Differential expression analysis of two groups was performed using the DESeq2 v.1.18.0 [33]. The *p*-values were adjusted using the Benjamini and Hochberg method. Genes with a *p*-value of less than 0.05 and log_2_ (fold change) larger than 1 were considered to be DEGs. Gene Ontology (GO) and Kyoto Encyclopedia of Genes and Genomes (KEGG) enrichment analysis of differentially expressed genes was implemented by the cluster Profiler v.3.8.1.

### 2.9. Quantitative RT-PCR

In order to verify the validity of the transcriptome data, 5 DEGs were selected for qRT-PCR with the PrimeScriptTM RT reagent Kit with gDNA Eraser (Perfect Real Time) (Takara Bio, Japan); the reaction was performed on the qTower3G Real-Time PCR System (Analytik Jena AG, Jena, Germany). The PCR amplification was initially heated to 95 °C, which lasted for 10 min, followed by, 40 cycles of 94 °C for 10 s, 60 °C for 20 s, and 60 °C for 20 s. The 16s was used as the internal reference gene to normalize gene expression, and the 2^-ΔΔCt^ method was used for relative quantification. The list of primers is given in Appendix A.

### 2.10. Statistical Analysis

Data were expressed as the mean ± standard error, and the data were subjected to a one-way analysis of variance (ANOVA) (*p* < 0.05) followed by a significant difference test (Tukey’s test) using SPSS statistics v21.0 (SPSS Inc., Chicago, IL, USA).

## 3. Results

### 3.1. Bioassay Results

The antibacterial activity of ethylicin against Psa in vitro was investigated via a turbi-dimeter test, which followed our reported methods [24]. As shown in Table 1, ethylicin performed excellently in all of the initial concentrations, with overwhelming suppression of Psa. Furthermore, a much lower gradient concentration from 0 to 5.00μg/mL was then set to evaluate its EC_50_ value. The EC_50_ value of ethylicin against Psa was 1.80 μg/mL, which was significantly better than the positive controls Kocide 3000 (EC_50_ = 37.72 μg/mL) and copper abietate (EC_50_ = 157.35 μg/mL). It was confirmed that ethylicin was more effective in combating Psa with remarkably higher activity than Kocide 3000 and copper abietate.

In vivo assay against kiwifruit bacterial canker was performed with a branch cut inoculation method on the one-year-old kiwifruit plant. In this trial, the blank control (with water treatment after cut but no inoculation) showed the normal growing of the kiwifruit plant with no infection. All of the other inoculated plants, over 14 days, exhibited canker symptoms with 100% morbidity (Figure 1 and Table 2). It can be seen that the lesion length (around 1 mm) of the kiwifruit branches treated with ethylicin was significantly shorter than that of the control. In addition, the inhibitory effect of Kocide 3000 (89.47%) was close to that of ethylicin (92.02%) at 500 μg/mL, but Kocide 3000 decreased to 89.47% at 200 μg/mL and was significantly lower than that of ethylicin (91.01%). The results indicate that pretreating kiwifruit plants with ethylicin can avoid the infestation of Psa even at a lower dosage when compared to Kocide 3000.

### 3.2. Psa Growth Concentration–Time Dependence on Ethylicin

Ethylicin inhibits the proliferation of Psa depending on concentration and time, which means that Psa growth was more obviously suppressed with the increase in ethylicin concentration; however, the Psa growth increased after a lag period but kept stasis when the concentration was extended to 7.20–9.0 μg/mL (Figure 2). The growth of Psa in 3.60 μg/mL ethylicin treatment showed a similar proliferation increasing tendency to that of the control group, and 48 h later they reached a final close level. When the content was over 3.60 μg/mL, the overwhelming effect of ethylicin on bacterial growth lasted for at least 24 h, along with basically no growth of Psa in the three treatment groups with higher concentrations: 4.50 μg/mL, 5.40 μg/mL, and 6.30 μg/mL. Moreover, 48 h later, the bacterial density with the treatment of 6.30 μg/mL fell to half that of the control group. The results showed that ethylicin can rapidly and significantly inhibit the growth of Psa and also act in a concentration- and time- dependent manner.

### 3.3. SEM and TEM Results

The effects of the different concentrations of ethylicin on the morphology of Psa were investigated by scanning electron microscopy, as shown in Figure 3. It can be seen that normal growth of Psa in the control group was even and uniform in shape (Figure 3a). Some of the Psa showed slight wrinkling when treated with 50 μg/mL of ethylicin (Figure 3b), and obvious wrinkling and rupture occurred in Psa cell membranes when ethylicin concentration increased to 75 μg/mL (Figure 3c) and 100 μg/mL (Figure 3d). Furthermore, almost whole clusters of Psa cells were actually cell fragments rather than complete cells.

The effect of ethylicin on Psa was also evaluated by TEM (Figure 4). It can be seen that the flagellum of Psa in 1.80 μg/mL of treatment grew normally like in the DMSO control, but when the ethylicin content was increased to 3.60 μg/mL (Figure 4c) and 5.40 μg/mL (Figure 4d), the flagellum was not observed, indicating that ethylicin at 3.60 μg/mL and 5.40 μg/mL started inhibiting the formation of Psa flagellum. This is corroborated with the transcriptome results, in which the flagellar formation pathway of Psa was inhibited in 3.60 μg/mL of ethylicin treatment, and the flagellar formation was not observed in Psa in 3.60 μg/mL of ethylicin treatment by transmission electron microscopy.

### 3.4. Effect of Ethylicin on Psa Biofilm Formation and Bacterial Migration

The ability of ethylicin to inhibit the formation of Psa biofilm was investigated, and the results are shown in Figure 5. Compared with the control, ethylicin could significantly inhibit the formation of Psa biofilm, and this inhibition effect became more obvious with the increase in ethylicin concentration, but when the concentration of ethylicin exceeded 2.70 μg/mL the inhibition of biofilm formation reached its peak and no longer increased.

To investigate the effect of ethylicin on bacterial motility, the swimming diameter of Psa is measured in the presence and absence of ethylicin. It can be seen in the figure that the diameter of the colonies is reduced by about 40% in the plates with 3.60 μg/mL of ethylicin, indicating that ethylicin significantly inhibited the motility of Psa (Figure 6).

### 3.5. Transcriptome Results

#### 3.5.1. Quality Control of Sequencing Data

Replicate samples of the control group (CK_1/2/3/4) and the ethylicin treatment group (ET_1/2/3/4) are included in this study. We obtained 7.61–7.96 million raw reads from the control group and 7.59–7.94 million raw reads from the ethylicin treatment group. After filtering and removing low-quality reads, total reads were 7.57–7.92 million and 7.54–7.89 million, respectively. Of these clean reads, the GC content was 53.04–54.76% and the Q30 values were over 94.31%. The ratio of total mapped reads of the control and ethylicin treatment groups was between 99.08–99.36% and 95.94–98.96% for Psa according to the Genome Database, which indicates that the quantity and quality of RNA-SEQ data were high. Unique mapped reads were 97.13–97.90% in the control group and 93.49–96.43% in the ethylicin treatment group (Appendix A).

#### 3.5.2. Differential Gene Expression Analysis

A total of 5389 differential genes were identified by transcriptome sequencing, using padj < 0.05, log_2_FoldChange > 0 as the standard. There were a total of 1793 genes with significant differences between the CK group and the ethylicn (ET) group, of which 954 differential genes were down-regulated and 839 genes were up-regulated (Figure 7 and Appendix A).

By clustering the differential gene expression values of the CK group and the ET group, the rows of expression data were normalized, and genes with similar expression patterns were clustered together to obtain a clustering heat map of differential genes between samples (Appendix A). The H-cluster method is used to divide the differential gene set into four clusters (Appendix A).

#### 3.5.3. Gene Ontology Annotation

We classified differentially expressed genes using Gene Ontology (GO) enrichment (Appendix A and Appendix A). Biological process (BP) enriched 746 differential genes into 230 pathways. The first three pathways with the most significant enrichment are amide biosynthetic process (GO: 0043604), organonitrogen compound biosynthetic process (GO: 1901566), and peptide biosynthetic process (GO: 0043043), indicating that the treatment of ethylicin is closely related to the biosynthesis of Psa (Appendix A). Cellular component (CC) enriched 264 differential genes into 35 pathways. The first three pathways with the most significant enrichment are intracellular (GO: 0005622), the cytoplasmic part (GO: 0044444), and the ribonucleoprotein complex (GO: 1990904) (Appendix A). Molecular function (MF) enriched 844 differential genes into 147 pathways. The first three pathways with the most significant enrichment are signal transducer activity (GO: 0004871), ligand activity (GO: 0016874), and oxidoreductase activity, acting on the CH-OH group of donors (GO:0016614) (Appendix A). Statistics of the first 10 pathway genes enriched in the three parts are shown in Figure 8.

#### 3.5.4. Kyoto Encyclopedia of Genes and Genomes (KEGG) Pathway Annotation

KEGG is a comprehensive database that integrates genomic, chemical, and system function information. The KEGG pathway analysis enriched 631 differential genes to 80 pathways (Appendix A), the 20 most significant KEGG pathways were selected for scatter plotting (Figure 9), and the gene changes of the top 10 most significant pathways were counted (Appendix A). The most significantly up-regulated pathway is protein export, with 11 genes up-regulated and no genes down-regulated, suggesting that ethylicin may accelerate protein secretion by Psa. The most significantly down-regulated pathway is flagellar assembly, with 20 genes down-regulated and 2 genes up-regulated, suggesting that ethylicin treatment may inhibit the flagellar assembly process of Psa, which in turn inhibits the formation and motility of Psa flagella. In particular, the pathway with significant up-regulation is also biofilm formation–*Pseudomonas aeruginosa*, with a total of 29 differential genes enriched, of which 20 are up-regulated and 9 are down-regulated. It is suggested that ethylicin may promote the up-regulated expression of genes associated with biofilm formation, which in turn promotes biofilm formation in Psa.

### 3.6. qRT-PCR Results

The effect of ethylicin on Psa was studied at the molecular level by qRT-PCR to analyze the expression of genes related to the bacterial virulence pathways. (Figure 10 and Appendix A). The Psa genes related to the biosynthesis of secondary metabolites (*gcvT* and *zwf*) are expressed at a lower basal level but are up-regulated (3–4-fold) in the presence of ethylicin. Interestingly, the Psa gene associated with the two-component system (*psTS* and *CN228_RS11330*) is strongly down-regulated (2–100-fold) in the presence of ethylicin. This gene may be involved in regulating bacterial flagella movement; *flgE* (regulating flagella biogenesis and coordinating flagella-dependent and flagella-independent movement) is down-regulated by 1.5-fold in the presence of ethylicin. The results show that the trend of qRT-PCR was consistent with the results of the RNA-Seq.

The results show that ethylicin can obviously inhibit the growth of Psa, disrupt the cell membrane of Psa, suppress its swimming movement, and the biofilm formation of Psa. The transcriptome results showed that ethylicin can up-regulate the expression of genes related to protein export and biofilm formation–*Pseudomonas aeruginosa*, and down-regulate the expression of genes related to flagellar assembly in Psa. Our results could help to gain a better understanding of the mechanisms by which ethylicin inhibits Psa.

## 4. Discussion

In this study, we evaluated the bactericidal activity of ethylicin against Psa and found that ethylicin exhibited high antibacterial activity against Psa with an EC_50_ value of 1.80 μg/mL. Ethylicin could significantly inhibit the growth of Psa with time–concentration dependence. For antibacterial mechanism analysis, SEM and TEM detection revealed that ethylicin could cause the Psa cell membrane to crumple and flagellar deformation.

It is necessary to mention that the result indicating that biofilm inhibition reached its peak at 2.70 μg/mL and no longer increased was consistent with the result of TEM, where the flagellar formation was inhibited at 3.60 and 5.40 μg/mL. It could be concluded that the flagellum assisted the adherence to the surface and differentiation into the biofilm. This phenomenon that the flagellum participated in the process of biofilm formation was proposed in many reports [34,35,36,37]. Hence, we conclude that ethylicin can damage the Psa cell membrane and flagellar formation, which avoid its biofilm formation and subsequently diminishes the Psa’s ability to infest the kiwifruit branches, thereby decreasing the occurrence of kiwifruit canker disease. Conducting transcriptome analysis on the possible mechanism of ethylicin on Psa and studying the changes in its related pathways and genes, we analyzed the results of Psa gene sequencing in the presence or absence of 3.60 μg/mL of ethylicin. Through KEGG, 631 genes were enriched into 80 pathways. The most down-regulated pathway was the flagellar assembly, and the most up-regulated pathway was biofilm formation–*Pseudomonas aeruginosa*.

The flagellar assembly pathway enriched 22 differential genes, of which, 20 differential genes are down-regulated, and 2 genes are up-regulated. We studied the effect of ethylicin on Psa swimming using the soft agar plate method and found that ethylicin at 3.60 and 7.20 μg/mL can significantly inhibit the movement of Psa, which coincided with the significantly down-regulated genes in the flagellar assembly pathway (*fliC*, *flgL*, *flgE*, and *flgH*) (Appendix A). The flagella movement of bacteria is essential for the colonization of bacteria in plant tissues, especially during the initial viscous contact process with plants [38]. Ethylicin can inhibit the movement of Psa, thereby inhibiting the virulence of Psa and reducing the risk of the kiwifruit infection of Psa.

In *Pseudomonas aeruginosa*, *fleQ* functions as a transcriptional regulator of expression of the exopolysaccharide Pel [39]. In *Pseudomonas putida*, disruption of *fleQ* caused strong defects in the biofilm formation, which relies on the supporting of exopolysaccharide [40]. A key component of biofilm formation in *Pseudomonas aeruginosa* is the biosynthesis of the exopolysaccharide Psl [41]. In the biofilm formation–*Pseudomonas aeruginosa* pathway, the up-regulated expression of *fleQ* genes promotes the up-regulated expression of Psl-related genes (Appendix A). It showed that Psa tries to form a biofilm to protect itself from the action of ethylicin. However, ethylicin did not promote the formation of a Psa biofilm, which may be related to the inhibition of Psa’s swimming movement. The formation of bacterial biofilm requires the participation of the flagella, and bacterial movement is necessary for the first stage of biofilm formation to reach the attachment surface [42]. As reported, Agrobacterium tumefaciens without the flagella cannot form biofilms [43]. The possible explanation is that ethylicin inhibits the flagellar assembly of Psa and reduces the swimming activity of Psa, thereby inhibiting the formation of a Psa biofilm.

## Figures and Tables

**Figure 1 microorganisms-09-00724-f001:**
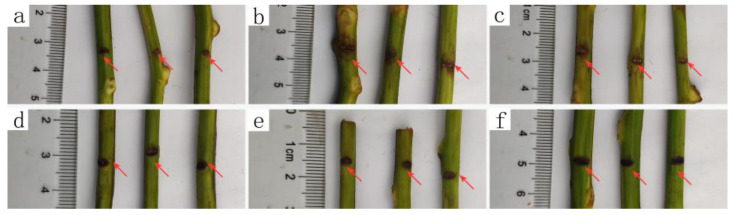
In vivo antibacterial activities of ethylicin against kiwifruit bacterial canker at 200 or 500 μg/mL on kiwifruit twigs: (**a**) blank control (no ethylicin and no bacterial inoculation); (**b**) Psa; (**c**) 200 μg/mL Kocide 3000; (**d**) 200 μg/mL ethylicin; (**e**) 500 μg/mL Kocide 3000; and (**f**) 500 μg/mL ethylicin.

**Figure 2 microorganisms-09-00724-f002:**
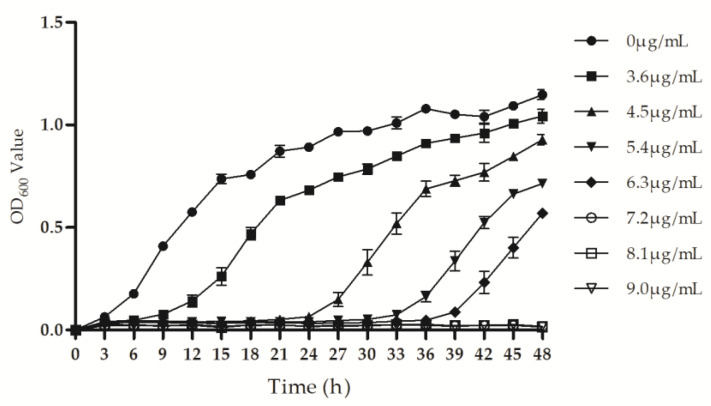
The 48 h growth curve of Psa treated with different concentrations of ethylicin.

**Figure 3 microorganisms-09-00724-f003:**
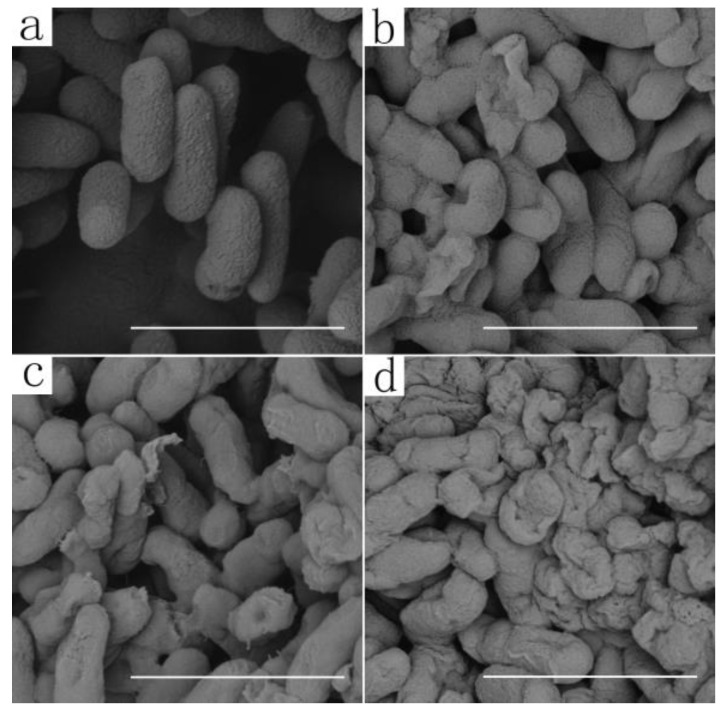
SEM images of Psa after being incubated with different concentrations of ethylicin: (**a**) 0 μg/mL; (**b**) 50 μg/mL; (**c**) 75 μg/mL; and (**d**) 100 μg/mL ethylicin. The scale bars for (**a**–**d**) are 2 μm.

**Figure 4 microorganisms-09-00724-f004:**
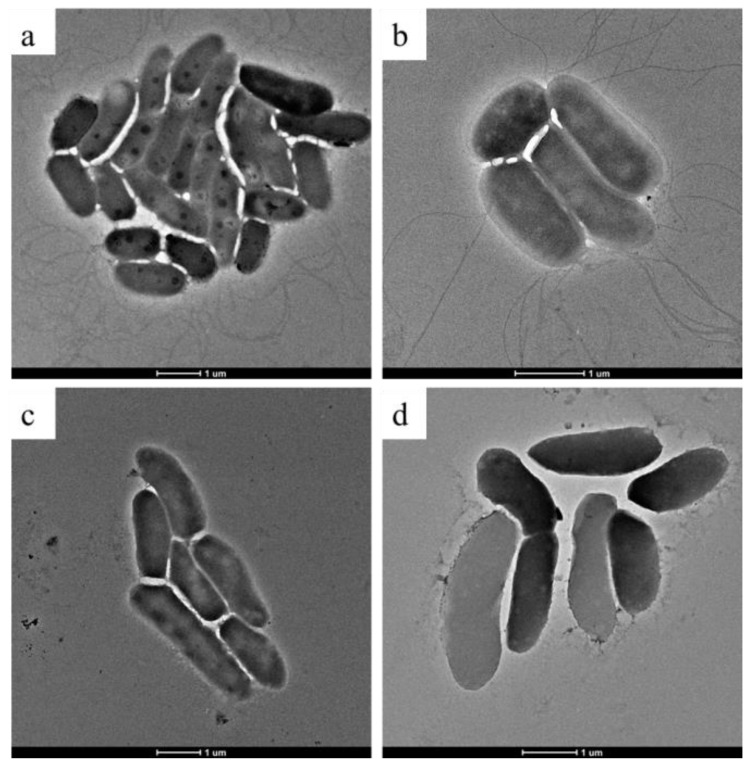
TEM images of Psa after incubation with different concentrations of ethylicin: (**a**) 0 μg/mL; (**b**) 1.80 μg/mL; (**c**) 3.60 μg/mL; and (**d**) 5.40 μg/mL ethylicin. The scale bars for (**a**–**d**) are 1 μm.

**Figure 5 microorganisms-09-00724-f005:**
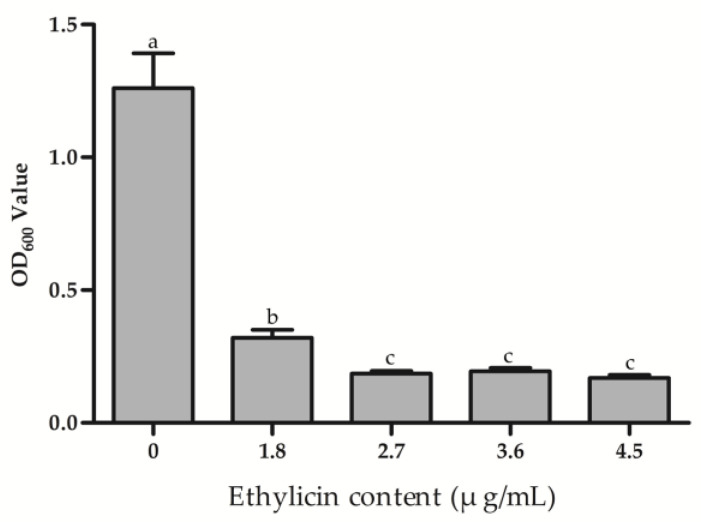
Biofilm formation content. The data represent the means ± SD of six replicate samples. The different letters (a–c) indicate significant differences at *p* < 0.05.

**Figure 6 microorganisms-09-00724-f006:**
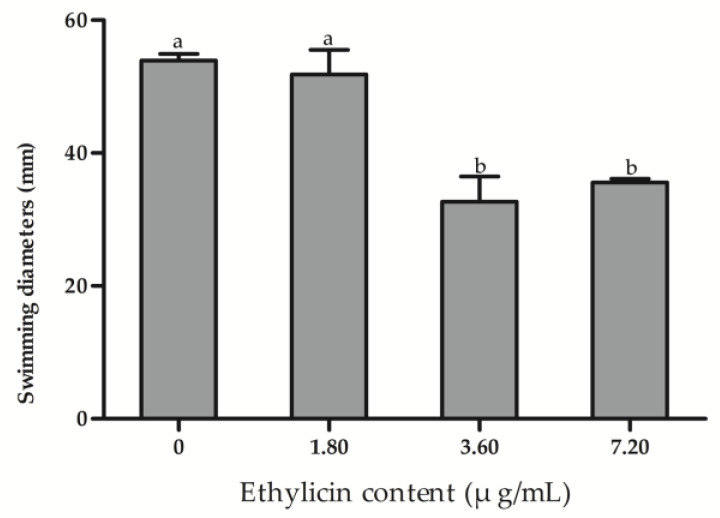
The measured value of the longest colony diameter. The data represent the means ± SD of three replicate samples. The different letters (a, b) indicate significant differences at *p* < 0.05.

**Figure 7 microorganisms-09-00724-f007:**
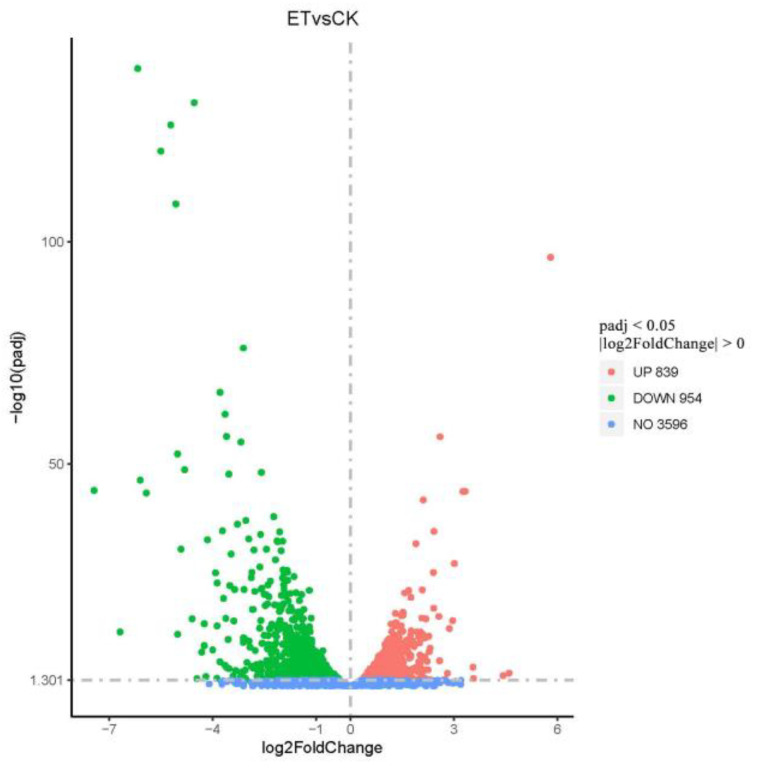
Volcano map of differentially expressed genes. Padj is the *p*-value after being corrected for multiple testing. The log2FoldChange is based on 2, and then the ratio of the gene expression levels of the treatment group and the control group is taken as the logarithmic value.

**Figure 8 microorganisms-09-00724-f008:**
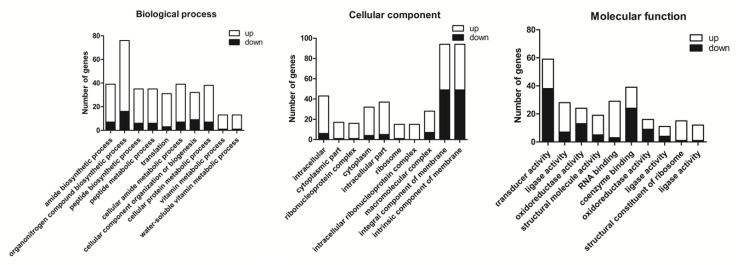
Biological process (BP), cellular component (CC), and molecular function (MF) enrichment of the number of genes in the top 10 pathways.

**Figure 9 microorganisms-09-00724-f009:**
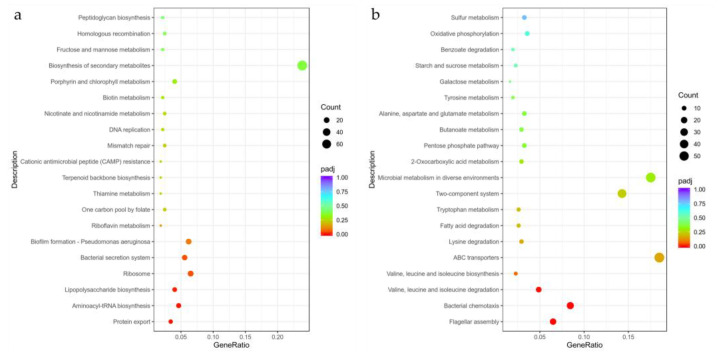
Kyoto Encyclopedia of Genes and Genomes (KEGG) enrichment analysis: (**a**) up-regulated; (**b**) down-regulated.

**Figure 10 microorganisms-09-00724-f010:**
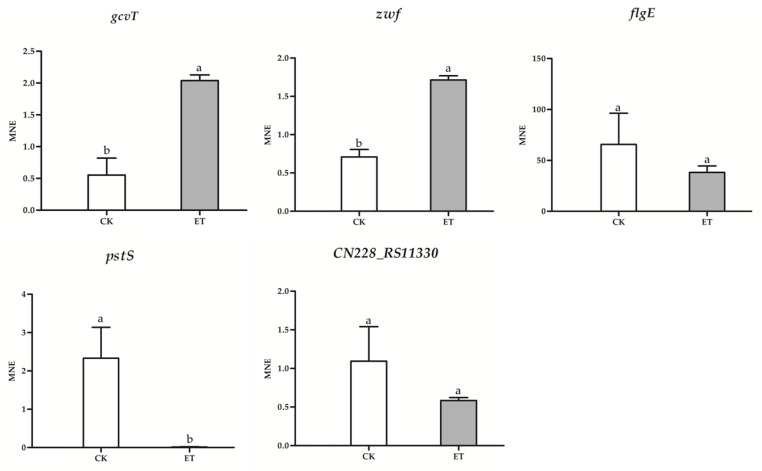
The qRT-PCR results. The data represent the means ± SD of three replicate samples (ET represent the average treatment of ethylicin at 3.6 μg/mL). Different letters (a, b) indicate significant differences at *p* < 0.05.

**Table 1 microorganisms-09-00724-t001:** In vitro antibacterial effects of three agents on *Pseudomonas syringae* pv. *actinidiae* (Psa).

Bactericide	Treatment (μg/mL)	Inhibition Rate (%)	Regression Equation	R	EC_50_ (μg/mL)
Ethylicin	50	99.59 ± 0.30	y = 3.4788x + 4.1133	0.9478	1.80 ± 0.06
	10	99.03 ± 0.17			
	5	98.46 ± 0.10			
Kocide 3000	100	94.60 ± 0.94	y = 3.9823x − 1.2784	0.9930	37.72 ± 2.12
	50	74.72 ± 3.24			
Copper abietate	100	44.15 ± 5.16	y = 0.8925x + 3.0393	0.9964	157.35 ± 5.88
	50	32.67 ± 2.22			

**Table 2 microorganisms-09-00724-t002:** Prevention effect of ethylicin on kiwifruit bacterial canker in vivo.

Treatment	14 Days after Inoculation		
	Morbidity (%)	Lesion Length (mm)	Control Efficiency (%)
(a) Blank control *	0	/	/
(b) Psa	100	6.27 ± 1.41a	/
(c) 200 μg/mL Kocide 3000	100	2.19 ± 0.87b	65.00% ± 13.92%b
(d) 200 μg/mL Ethylicin	100	0.56 ± 0.24 b	91.01% ± 3.87% a
(e) 500 μg/mL Kocide 3000	100	0.66 ± 0.26b	89.47% ± 4.10%a
(f) 500 μg/mL Ethylicin	100	0.5 ± 0.16 b	92.02% ± 2.51% a

* Blank control (no ethylicin and no bacterial inoculation). The different letters (a, b) indicate significant differences.

## Data Availability

The data presented in this study are available in Appendix A.

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
