# Peer review of "Transcriptome Analysis on the Mechanism of Ethylicin Inhibiting Pseudomonas syringae pv. actinidiae on Kiwifruit"

_microorganisms, 2021, doi:10.3390/microorganisms9040724_

Round 1

Reviewer 1 Report

These are my comments about the manuscript:

  1. Line 21 the word “understand” to be changed to “understanding”
  2. In line 16 the word “migration” need to be changed to “movement”
  3. In line 30 the word “discovered” need to be changed to “reported”
  4. In line 42 the words “applied compromised” not clear to me what they what to say. Need change.
  5. Line 64. There is no need to mention composition of Nutrient broth as it is an old and well- known media everywhere.
  6. Line 59 they need to report the source for ethylicin just like the other two pesticides.
  7. Line 69 it is not clear or the authors did not explain why they used DMSO as blank. Secondly, they need to spell out the word especially when ued for the first time.
  8. Line 72 “with reference” should be changed to “following the method of ---“.
  9. Lines 72 and 73 it is not clear to me if they used branches that has been cut from the trees or the branches were still on the tree. Need clarification.
  10. Line 73 the word “treated” need to change to “injected”.
  11. Line 81 C and T average length of the lesions in what unit? Need to write perhaps mm.
  12. Line 84 “by referring to”  need to be changed to “following”.
  13. In Line 107 instead of plates (0.3%) agar powder), should be said plates of 0.3% sterile water agar. There is no need to the word powder. Almost everything is powder in microbiology.
  14. Line 113- 122 the method is not clear. For example, after removing the culture, they say the wells were washed three times. Did they collect the wash water? This should be explained. Then they say “left” in the well. Instead of left should say “remaining”. Also, to me when you wash three times nothing should be left unless you perhaps say “rinsed three times gently.”
  15. Line 120 the word “feezed” is wrong. Perhaps they wanted to say frozen.
  16. Line 132 “was checked” change to “was evaluated.”
  17. Line 131 “tracking” change to “following”.
  18. In Results Table 1 need to write OD in a new column for each chemical and need to explain if the three chemicals were used at the same conc. Also, they did not mention anything about EC50 in the methods. EC50 should be clearly defined in the methods.
  19. Lines 182,183 the Figures to me it is not clear what they measured. Please put arrows to areas you measured.
  20. Line 192 “got aroused” to be changed to “increased after a long lag period”. They should make it clear that the chemical increased the lag period.
  21. Fig 2 near line 200, there is no label for x axis. Also, the Figure shows that none of the concentrations used inhibited growth completely. Therefore, they should have increased concentration to see if 100% inhibition was possible.
  22. SEM image Lines 210-211, Fig 3 better to have a picture with less dense of bacteria.
  23. Line 231 There are a lot of Figures in this section and I believe most can be in supplement section.

Author Response

Responses to the Reviewer 1

Reviewer’s comment:  Line 21 the word “understand” to be changed to “understanding”.

Our responses: Yes, and it is done according to your suggestions.

Reviewer’s comment:    In line 16 the word “migration” need to be changed to “movement”.

Our responses: Yes and it is done according to your suggestions.

Reviewer’s comment:    In line 30 the word “discovered” need to be changed to “reported”

Our responses: Yes, and it is done according to your suggestions.

Reviewer’s comment: In line 42 the words “applied compromised” not clear to me what they what to say. Need change.

Our responses: Thanks, this sentence was revised as “Sulfur agent was applied but with low efficiency (EC50,1326.99 mg/L) on Psd, which is but compromising to the concept of high-efficient safe pesticides”.

Reviewer’s comment: There is no need to mention composition of Nutrient broth as it is an old and well- known media everywhere.

Our responses: We delete the description about composition of Nutrient broth according to your suggestions.

Reviewer’s comment: Line 59 they need to report the source for ethylicin just like the other two pesticides.

Our responses: Actually both 80% ethylicin and 12% copper rosinate were purchased from Hainan Zhengye. But the parenthesis seemed did not cover it. So, this part was revised as “The pesticides used in the experiment were commercially available. Both 80% ethylicin and 12% copper rosinate were obtained from Hainan Zhengye Zhongnong High-tech Co., Ltd., Haikou, Hainan Province, China”

Reviewer’s comment: Line 69 it is not clear or the authors did not explain why they used DMSO as blank. Secondly, they need to spell out the word especially when ued for the first time.

Our responses: The reason for using DMSO as control is to subtract the interference of DMSO itself on the Psa. And it would be clear to name the group as negative control other than blank control.  And it was revised as “Sterile water containing dimethyl sulfoxide (DMSO) was used as negative check.”.

Reviewer’s comment: Line 72 “with reference” should be changed to “following the method of ---“.

Our responses: It has been revised according to your suggestions.

Reviewer’s comment: Lines 72 and 73 it is not clear to me if they used branches that has been cut from the trees or the branches were still on the tree. Need clarification.

Our responses: We performed the experiment on the living plant but observed it by cutting it off. The manuscript has been revised accordingly.

Reviewer’s comment: Line 73 the word “treated” need to change to “injected”.

Our responses: It has been revised according to your suggestions.

Reviewer’s comment: Line 81 C and T average length of the lesions in what unit? Need to write perhaps mm.

Our responses: “mm” was added according to your suggestion.

Reviewer’s comment: Line 84 “by referring to” need to be changed to “following”.

Our responses: It has been revised according to your suggestions.

Reviewer’s comment: In Line 107 instead of plates (0.3%) agar powder), should be said plates of 0.3% sterile water agar. There is no need to the word powder. Almost everything is powder in microbiology.

Our responses: It has been revised according to your suggestions.

Reviewer’s comment: Line 113- 122 the method is not clear. For example, after removing the culture, they say the wells were washed three times. Did they collect the wash water? This should be explained. Then they say “left” in the well. Instead of left should say “remaining”. Also, to me when you wash three times nothing should be left unless you perhaps say, “rinsed three times gently.”

Our responses: We are sorry for the vague description. We do not collect the wash water but check what remaining in the well after rinsing. We changed the expression according to your suggestions.

Reviewer’s comment: Line 120 the word “feezed” is wrong. Perhaps they wanted to say frozen.

Our responses: It was changed as frozen.

Reviewer’s comment: Line 132 “was checked” change to “was evaluated.”

Our responses: It is done according to your suggestions.

Reviewer’s comment: Line 131 “tracking” change to “following”.

Our responses: It is done according to your suggestions.

Reviewer’s comment: In Results Table 1 need to write OD in a new column for each chemical and need to explain if the three chemicals were used at the same conc. Also, they did not mention anything about EC50 in the methods. EC50 should be clearly defined in the methods.

Our responses: We have initially test all the agents at the same concentration. But there is no activtity gap for Ethylicin which indicated almost 100% inhibition rate at all the test level (from 200 to 5 ug/mL). Here in the revision, according to your suggestion we add concentration for comparison for three agents.

Table 1. In vitro antibacterial effects of three compounds on Psa.

Compd.

treatment (μg/mL)

Inhibition rate (%)

regression equation

R

EC50 (μg/mL)

Ethylicin

50

99.59±0.30

y=3.4788x+4.1133

0.9478

1.80±0.06

10

99.03±0.17

5

98.46±0.10

Kocide 3000

100

94.60±0.94

y=3.9823x-1.2784

0.9930

37.72±2.12

50

74.72±3.24

Copper abietate

100

44.15±5.16

y=0.8925x+3.0393

0.9964

157.35±5.88

50

32.67±2.22

Reviewer’s comment: Lines 182,183 the Figures to me it is not clear what they measured. Please put arrows to areas you measured.

Our responses: We put arrows to the measured areas. The manuscript has been revised according to your suggestions.

Reviewer’s comment: Line 192 “got aroused” to be changed to “increased after a long lag period”. They should make it clear that the chemical increased the lag period.

Our responses: This part has been revised according to your suggestions.

Reviewer’s comment:    Fig 2 near line 200, there is no label for x axis. Also, the Figure shows that none of the concentrations used inhibited growth completely. Therefore, they should have increased concentration to see if 100% inhibition was possible.

Our responses: We carried out new experiment by extending to higher concentrations from 7.2-90 and find out the Psa was all completely inhibited (100% inhibition rate).

Reviewer’s comment: SEM image Lines 210-211, Fig 3 better to have a picture with less dense of bacteria.

Our responses: We listed the original picture obtained from the SEM graph. And in the revised manuscript a new TEM (Transmission Electron Microscopy) image was added with less dense of bacteria.

And some new references were insert and also added and the number of reference list were adjusted.

Reviewer’s comment: Line 231 There are a lot of Figures in this section and I believe most can be in supplement section.

Our responses: The manuscript has been revised according to your suggestions. We put Table 3 and Fig 7,8,9 in supporting information and referred the name of figure and table as “Table S1, 2.., and FigS1,2, 3..” in the revision.

Reviewer 2 Report

The manuscript must be improved in

Author Response

Responses to the Reviewer 2

Reviewer’s comment: The authors described the effect of ethylicin on Pseudomonas syringae pv. actinidiae growing and the prevention in the cankering the kiwifruit stem. In addition, the authors perform a transcriptomics research to determine ethylicin-regulated up and down genes. The English language and grammar should be strongly improved.

Our responses: Thanks for your comments and suggestions on our manuscript. We have polished the English writing and revised the manuscript.

Reviewer’s comment: Introduction: Review English language and add more info on ethylicin.

Our responses: According to your suggestions, we provided the chemical information about ethylicin as “The Organosulfur compound ethylicin (S-Ethyl ethanethiosulfonate; CAS number 682-91-7) “

Reviewer’s comment: Materials and Methods: In general, authors should better describe the Materials and Methods section, especially paragraphs 2.3-2.5 and 2.6.

Our responses: We described the method more precisely and added and a TEM (Transmission Electron Microscopy) experiment with references. The manuscript has been revised according to your suggestions.

Reviewer’s comment: Line 59: …80% ethylicin… Where ethylicin come from? Please, add this information.

Our responses: Actually both 80% ethylicin and 12% copper rosinate were purchased from Hainan Zhengye. It is revised according to your suggestions.

Reviewer’s comment: Line 89: What tool did the authors use to measure OD600? Please, add this information.

Our responses: The manuscript has been revised according to your suggestions. “The growth of the cultures was monitored on a microplate reader (Synergy H1, Bio Tek, USA)

Reviewer’s comment: Line 107: …agar plate…What kind of agar plate? Nutritive or selective agar? Please, add this information and also add the medium composition.

Our responses: The agar plate is Nutritive agar with 3% ager in normal NB medium(10 g glucose, 5 g, peptone, 1 g yeast powder, 3 g beef extract dissolved in 1000 mL deionized water, pH=7.0-7.2.)  The manuscript has been revised  as “…were transferred on 0.3% agar with nutrient broth medium .

Reviewer’s comment: Results: Bioassay results: Why did the authors not perform the in vivo assay using Kocide 3000 as a comparison with ethylicin?

Our responses: Since ethylicin performed far better than the control agents on Psa in vitro so we did not apply it in the following in vivo assay.  

But in our continuous work we did compare the activity of Kocide 3000 and we provided the new data in the revised manuscript.

Table 2. Prevention effect of Ethylicin on Kiwifruit bacterial canker in vivo.

Treatment

14 Days after Inoculation

Morbidity (%)

Lesion length (mm)

Control Efficiency (%)

a) Blank control*

0

/

/

b) Psa

100

6.27±1.41a

/

c)200 μg/mL Kocide 3000

100

2.19±0.87b

65.00%±13.92%b

d)200 μg/mL Ethylicin

100

0.56±0.24 b

91.01%±3.87% a

e)500 μg/mL Kocide 3000

100

0.66±0.26b

89.47%±4.10%a

f)500 μg/mL Ethylicin

100

0.5±0.16 b

92.02%±2.51% a

*Blank control (no ethylicin, no bacterial inoculation).

Reviewer’s comment: SEM results: The authors could use SEM to investigate the morphology of Psa when treated with 6.3µg/mL ethylicin, as it is the concentration that performed best in the inhibition of the Psa proliferation.

Our responses: We did examine the morphology of Psa at 1.8, 5.4, 9.0 µg/mL but there was no obvious disform and those data and figure were not given in the MS. 

But the additional experiment TEM (Transmission Electron Microscopy) provided the morphology of Psa treated by 1.8 μg/mL, 3.6 μg/mL and 5.4 μg/mL and 9.0 µg/mL.

SEM images for Psa after incubated with different concentrations of ethylicin. (a) 0 μg/mL, (b) 1.8 μg/mL, (c) 5.4 μg/mL and (d) 9.0 μg/mL ethylicin. Scale bars for a,b,c,d are 2 μm.

Reviewer’s comment: Lines 216-219: Authors should give their own interpretation of why if ethylicin exceeded 2.7µg/mL, the inhibition reached the peak and no longer increased.

Our responses: We have tried to explain this phenomenon in the discussion section as following “It was necessary to mention that the result that biofilm inhibition reached the peak at 2.7 μg/mL and no longer increased was consist with the result of TEM where the flagellar formation was inhibited at 3.6 and 5.4 μg/mL. This might be concluded that flagellar assist adherence to the surface and differentiation into the biofilm. This phenomenon that the flagellar participate the process of biofilm formation was proposed in many reports[34-37].”

  • Duan, Q.D.; Zhou, M.X.; Zhu, L.Q.; Zhu; G.Q. Flagella and bacterial pathogenicity. Basic Microb. 2012, 52, 1-8.
  • Chaban, B.; Hughes, H.V.; Beeby M. The flagellum in bacterial pathogens: For motility and a whole lot more. Cell Dev. Biol. 2015, 46, 91-103.
  • Lemon, K.P.; Higgins, D.E.; Kolter, R. Flagellar motility is critical for Listeria monocytogenes biofilm formation. Bacteriol. 2007, 189, 4418-4424.
  • Pratt, L.A.; Kolter, R. Genetic analysis of Escherichia coli biofilm formation: roles of flagella, motility, chemotaxis and type I pili. Microbiol. 1998, 30, 285-293.

Reviewer’s comment: Figure 11: Please, increase resolution.

Our responses: The resolution of the KEGG image has been adjusted to above 300dpi.

Reviewer’s comment: Section 3.6 qRT-PCR results: The activity of ethylicin on pstS and CN228-RS11330 genes regulation is not described by the authors. Please insert these informations.

Our responses: It’s done according to your suggestions.

Reviewer’s comment: Discussion: Lines 338-360: The authors described the effect of ethylicin on Psa swimming which coincides to the down-regulated genes fliC, flgL, flgE, flgH and flgQ. In the manuscript, the authors described only the down-regulation of gene flgE. Please insert the results about fliC, flgL, flgH and flgQ genes down-regulated.

Our responses: The other down-regulated fliC, flgL, flgH and fleQ genes and up-regulated fleQ were also illustrated in the following figure in our present study. In the ET group, the down-regulated expression of flgH was 3.75 fold, the down-regulated expression of flgE was 3.47 fold, the down-regulated expression of fliC was 3.84 fold, the down-regulated expression of flgL was 2.96 fold, and the up-regulated expression of flgQ was 1.41 fold.

Figure a Flagellar assembly pathway, Figure b Biofilm formation-Pseudomonas aeruginosa pathway

  Table1 five differential gene expression

gene_id

gene_name

ETvsCK_log2FoldChange

gene_description

CN228_RS21750

flgH

-1.907

flagellar basal body L-ring protein FlgH && PF02107:Flagellar L-ring protein

CN228_RS21775

flgE

-1.797

flagellar hook protein FlgE && PF00460:Flagella basal body rod protein|PF07559:Flagellar basal body protein FlaE|PF06429:Flagellar basal body rod FlgEFG protein C-terminal

CN228_RS21710

fliC

-1.942

flagellin && PF00700:Bacterial flagellin C-terminal helical region|PF00669:Bacterial flagellin N-terminal helical region

CN228_RS21730

flgL

-1.568

flagellar hook-associated protein 3 && PF00669:Bacterial flagellin N-terminal helical region

CN228_RS21685

FleQ

0.498

sigma-54-dependent Fis family transcriptional regulator && PF06490:Flagellar regulatory protein FleQ|PF02954:Bacterial regulatory protein, Fis family|PF00158:Sigma-54 interaction domain

Round 2

Reviewer 2 Report

The authors responded to comments. The English language can be further improved.